# Management of Resectable Stage III-N2 Non-Small-Cell Lung Cancer (NSCLC) in the Age of Immunotherapy

**DOI:** 10.3390/cancers13194811

**Published:** 2021-09-26

**Authors:** Xabier Mielgo-Rubio, Sara Montemuiño, Unai Jiménez, Javier Luna, Ana Cardeña, Laura Mezquita, Margarita Martín, Felipe Couñago

**Affiliations:** 1Department of Medical Oncology, Hospital Universitario Fundación Alcorcón, 28922 Madrid, Spain; a.cardenna@gmail.com; 2Department of Radiation Oncology, Hospital Universitario Fuenlabrada, 28942 Madrid, Spain; sara.montemuino@salud.madrid.org; 3Department of Thoracic Surgery, Hospital Universitario Cruces, 48903 Barakaldo, Bizkaia, Spain; unai.jimenezmaestre@osakidetza.eus; 4Department of Radiation Oncology, Fundación Jiménez Díaz, 28040 Madrid, Spain; jluna@fjd.es; 5Department of Medical Oncology, Hospital Universitari Clínic Barcelona, 08036 Barcelona, Spain; lmezquita@clinic.cat; 6Department of Radiation Oncology, Hospital Universitario Ramón y Cajal, 28034 Madrid, Spain; margarita.martin@salud.madrid.org; 7Department of Radiation Oncology, Hospital Universitario Quirónsalud Madrid, 28223 Madrid, Spain; felipe.counago@quironsalud.es; 8Department of Radiation Oncology, Hospital La Luz, 28003 Madrid, Spain; 9Medicine Department, School of Biomedical Siciences, Universidad Europea, 28670 Madrid, Spain

**Keywords:** non-small-cell lung cancer, NSCLC, N2, resectable, immunotherapy, stage III, perioperative

## Abstract

**Simple Summary:**

The treatment of resectable stage III non-small-cell lung cancer with N2 lymph node involvement is usually multimodal and is generally based on neoadjuvant chemotherapy +/− radiotherapy followed by surgery, but the cure rate is still low. Immunotherapy based on anti-PD1/PD-L1 immune checkpoint inhibitors has improved survival in advanced and stage III non-resectable NSCLC patients and is being studied in earlier stages to improve the cure rate of lung cancer. In this article, we review all therapeutic approaches to stage III-N2 NSCLC, analysing both completed and ongoing studies that evaluate the addition of immunotherapy with or without chemotherapy and/or radiotherapy.

**Abstract:**

Stage III non-small-cell lung cancer (NSCLC) with N2 lymph node involvement is a heterogeneous group with different potential therapeutic approaches. Patients with potentially resectable III-N2 NSCLC are those who are considered to be able to receive a multimodality treatment that includes tumour resection after neoadjuvant therapy. Current treatment for these patients is based on neoadjuvant chemotherapy +/− radiotherapy followed by surgery and subsequent assessment for adjuvant chemotherapy and/or radiotherapy. In addition, some selected III-N2 patients could receive upfront surgery or pathologic N2 incidental involvement can be found *a posteriori* during analysis of the surgical specimen. The standard treatment for these patients is adjuvant chemotherapy and evaluation for complementary radiotherapy. Despite being a locally advanced stage, the cure rate for these patients continues to be low, with a broad improvement margin. The most immediate hope for improving survival data and curing these patients relies on integrating immunotherapy into perioperative treatment. Immunotherapy based on anti-PD1/PD-L1 immune checkpoint inhibitors is already a standard treatment in stage III unresectable and advanced NSCLC. Data from the first phase II studies in monotherapy neoadjuvant therapy and, in particular, in combination with chemotherapy, are highly promising, with impressive improved and complete pathological response rates. Despite the lack of confirmatory data from phase III trials and long-term survival data, and in spite of various unresolved questions, immunotherapy will soon be incorporated into the armamentarium for treating stage III-N2 NSCLC. In this article, we review all therapeutic approaches to stage III-N2 NSCLC, analysing both completed and ongoing studies that evaluate the addition of immunotherapy with or without chemotherapy and/or radiotherapy.

## 1. Introduction

Lung cancer is the most common type of cancer worldwide, with 2.1 million new cases per year. Non-small-cell lung carcinoma (NSCLC) is the most common histological subtype (≈85%) [1]. While in recent years, there seems to be a trend of an increasing number of patients being diagnosed in the localised stage (from 16.6% in 1988 to 23.6% in 2015, according to the SEER database), most patients continue to be diagnosed in the advanced age [2]. According to the 2019 National Lung Cancer Audit annual report from the United Kingdom, only 22% of lung tumours are diagnosed in stage III, with 10% patients being diagnosed in stage IIIA [3].

Stage III NSCLC patients are a highly heterogeneous group, as can be seen in the TNM classification system, with different tumour sizes and different degrees of lymph node involvement. This means these patients may be candidates for different therapeutic approaches, either with initial surgery, neoadjuvant therapy followed by surgery or non-surgical multimodality treatment, including radical treatment with radiotherapy. Stage III NSCLC with N2 lymph node involvement includes those patients with ipsilateral mediastinal and/or subcarinal lymph node metastases involvement. The TNM classification system has not varied between the 7th and the last 8th edition, sub-classifying tumours into Stage IIIA and IIIB depending on the size and degree of infiltration of the primary tumour: IIIA (T1-2 N2) and IIIB (T3-4 N2) [4]. In practical terms, stage III-N2 NSCLC cases can also be classified as resectable or unresectable. Unresectable patients are those who, even following induction treatment, cannot obtain complete microscopic resection (R0), and tend to be those with bulky N2 involvement and those with T4 involvement (with diaphragm, mediastinal, heart, large vessel, trachea, recurrent laryngeal nerve, oesophagus, vertebral body, or carina infiltration or with tumour nodules separated in different lobes of the same lung). Nevertheless, we must always evaluate resectability within a multidisciplinary committee. In this review, we focus on the management of those patients with resectable stage III-N2 NSCLC, or in other words, those patients who can opt for initial surgery or are potentially resectable with surgery that does not require pneumonectomy following prior neoadjuvant treatment.

Despite the advances in the lung cancer field in recent years, the prognosis for these tumours is still fairly poor and has ample room for improvement, with a 5-year overall survival (OS) rate of 36% and 26% for stages IIIA and IIIB, respectively [5]. Immunotherapy has revolutionised NSCLC treatment and is now considered an established treatment that has improved survival rates and quality of life for patients with advanced and unresectable locally advanced disease [6,7]. The next step will be to incorporate immunotherapy into treatment for resectable or potentially resectable early stages. In this manuscript, we review how the arrival of immunotherapy represents a paradigm change for the management of patients with resectable stage III-N2 NSCLC, and we perform a critical review of the studies that have evaluated or currently are evaluating immunotherapy within this context.

## 2. Current Situation, Guidelines and Clinical Practice

According to the clinical guidelines of the National Comprehensive Cancer Network (NCCN) version 2.2021, non-invasive N2 cT1-2, T3 tumours can be treated with induction chemotherapy (ChT) with subsequent surgical rescue or concomitant chemo-radiotherapy (CRT) with response-dependent radical intent, or immediately with radical concomitant CRT followed by consolidation durvalumab according to the indications in each country (only approved in PD-L1 ≥ 1% by the European Medicines Agency). On the other hand, in invasive cT3N2 tumours and cT4N2, concomitant CRT with radical intent followed by consolidation durvalumab is suggested as the treatment of choice [8].

The guidelines from the Spanish Society of Medical Oncology (SEOM) from 2019 differentiate between three situations [9]:Incidental IIIA-N2: N2 involvement is observed during surgery or during analysis of the surgical specimen, in which adjuvant ChT with four cycles of platinum-based doublet is recommended, and the subsequent evaluation of postoperative radiotherapy (PORT) in those patients with free margins. In cases with affected margins, complementary radiotherapy (RT) should be started followed by adjuvant ChT.Potentially resectable IIIA-N2: start neoadjuvant induction ChT, generally with three cycles of platinum-based doublet, with subsequent re-evaluation and staging and, in the case of response with downstaging, complete resection surgery with lymphadenectomy and subsequent assessment for adjuvant ChT. There are studies and some centres that have treated patients with neoadjuvant CRT, but this strategy requires very close coordination between the medical oncology, radiation oncology and thoracic surgery services and has not been shown to increase survival rates.Unresectable IIIA-N2: immediate CRT treatment with radical intent followed by assessment for consolidation durvalumab.

The most recent version of the clinical guidelines from the European Society of Medical Oncology (ESMO) suggest management similar to that suggested by the SEOM: adjuvant ChT in the case of incidental N2, multimodality treatment that includes resection surgery in potentially resectable N2 cases and non-surgical multimodality treatment in those with unresectable N2 disease [7].

Regarding adjuvant ChT treatment in those with incidental N2 involvement, the standard is four cycles of adjuvant cisplatin-based doublet, after having demonstrated an increase in OS at 5 years of 5.4% and a reduction in the risk of death of 11% (HR 0.89, 95%CI 0.82–0.96, *p* = 0.0043) in the LACE meta-analysis, which included five studies (ANITA, JBR.10, BLT, IALT and ALPI) with a total of 4584 patients [10]. The ChT schedules with the most accumulated evidence are cisplatin + vinorelbine and cisplatin + etoposide, though the latter is rarely used in clinical practice, as it is more toxic.

In patients with potentially resectable Stage III-N2, neoadjuvant ChT with platinum-based doublets treatment is usually suggested, which has been shown to have the same impact on improving OS as adjuvant treatment, though with slightly less evidence [11,12,13,14,15,16]. In general, the recommendation is to administer three cycles of ChT followed by re-evaluation through imaging and mediastinal lymph node staging.

## 3. Role of Pre- and Post-Surgical Radiotherapy in III-N2 NSCLC

The addition of RT as part of neoadjuvant treatment for improving local control was explored, which could impact survival results. Various recent meta-analyses confirm the absence of the impact of neoadjuvant CRT on survival rates [17,18], but it does obtain a higher rate of local control, downstaging and complete resection [19,20,21,22,23,24].

Dose increases in RT above the established standard (45–50 Gy), as well as accelerated RT schedules, showed a significant increase in mediastinal negativisation (75–90%), with a higher percentage of pathologic complete response (pCR) versus the RT at standard doses without differences in morbidity and mortality [25,26,27,28,29,30].

Regarding PORT, its role in preventing local recurrence in IIIA N2 NSCLC patients with incomplete surgical resection is known, and significantly improved survival in all the subgroups of lymph node disease [31]. In completely resected patients, PORT has been a topic of debate since the publication of the PORT Meta-analysis Trialist Group [32], as it is associated with being a detriment to survival by increasing the risk of death by 21%. The studies analysed in this meta-analysis used obsolete techniques and treatment schedules, which are possibly responsible for cardiac and pulmonary toxicity.

The ANITA study, which evaluated adjuvant ChT in patients with resected NSCLC, observed that the pN2 patient subgroup that received ChT and PORT presented a benefit in OS versus the observation (5-y OS: 47.4% vs. 34%) [33]. Later, multiple studies compared adjuvant ChT treatment schedules followed by PORT versus exclusive ChT and confirmed the benefit to survival with the use of modern RT techniques, producing OS rates at 5 years of around 35%, with a significant benefit to those patients receiving between 45 and 54 Gy (HR 0.85, *p* < 0.001) [34,35,36,37]. This marked a turning point for the inclusion of PORT within the clinical guidelines for N2 patients.

Recently, the randomised phase III LungART study analysed the value of PORT in patients with completely resected stage IIIA-N2 NSCLC that received standard ChT. In total, 501 patients were randomised to receive PORT (54 Gy in 27–30 fractions) vs. observation, achieving a 3-year disease-free survival rate (DFS) and 3-year OS of 47.1% vs. 43.8% and 66.5% vs. 68.5%, respectively, without finding statistically significant differences but with a higher death risk in the PORT arm mainly due to heart and lung toxicity [38]. Another randomised phase III study, published in 2021, evaluated PORT (50 Gy) in Chinese patients with totally resected IIIA-N2 NSCLC and found no differences in DFS (HR 0.84, 95%CI 0.65–1.09; *p* = 0.20) despite achieving a better local recurrence-free survival (HR 0.71, 95%CI 0.51–0.97; *p* = 0.03) [39]. These data bring down the results of previous studies, once again casting doubt on the role of PORT in completely resected patients. Therefore, more studies are needed to identify which patient subgroups PORT could reduce the rate of local relapse in, thus improving survival.

## 4. Immunotherapy in Adjuvant Setting in Resected Stage III-N2 NSCLC

Initially, the immunotherapy strategy that started to be studied in NSCLC within the context of adjuvant treatment were vaccines against the MAGE-A3 antigen, with no differences in DFS (60.5 vs. 57.9 mo, HR 1.02) found in a phase III trial [40].

Later, immune checkpoint inhibitors (ICIs) and anti-interleukin-1β (IL1β) antibodies began to be studied as adjuvant treatments, either as monotherapy, in combination with other ICIs (double immunotherapy) or in combination with ChT. However, all these studies are ongoing, and even though they have already included more than 5000 patients, results are not available (Table 1).

### 4.1. Adjuvant Treatment with Immunotherapy in Monotherapy

There are currently six phase III clinical trials evaluating the role of adjuvant mono-immunotherapy in patients with completely resected III-N2 NSCLC, with five of which evaluate anti-PD1 or anti-PD-L1 ICIs (PEARLS, BR31, ANVIL, IMpower010, ALCHEMIST-IO Arm B), including operated patients with stage IIIA-N2 and excluding T3-4N2 tumours, and one of them evaluating the anti-IL1β monoclonal antibody canakinumab in both IIIB-N2 as well as IIIA-N2 patients (CANOPY-A) [41,42,43,44]. All these studies use the adjuvant immunotherapy strategy during 1 full year following surgery and standard adjuvant treatment, except for one arm of the ALCHEMIST-IO (Arm B), which uses 17 cycles of pembrolizumab. The primary aim of all the studies is DFS, while the ANVIL and ALCHEMIST-IO studies also have OS as a co-primary objective [45]. The PEARLS study, which evaluates adjuvant pembrolizumab, and the BR31 study, which evaluates adjuvant durvalumab, as well as CANOPY-A, which evaluates adjuvant canakinumab, are randomised studies with placebo; the ANVIL study, evaluating adjuvant nivolumab, the IMpower010 study, evaluating adjuvant atezolizumab, and the ALCHEMIST Chemo-IO study, evaluating adjuvant pembrolizumab, are randomised studies with observation. Among these studies, Impower 010 was the first one to report primary results in the American Society of Clinical Oncology (ASCO) Annual Meeting 2021, showing a DFS benefit with adjuvant atezolizumab after adjuvant chemotherapy in patients with completely resected stage II-IIIA NSCLC, with pronounced benefit in patients with PD-L1 expression ≥ 1% [46].

### 4.2. Adjuvant Treatment with Immunotherapy in Combination with Chemotherapy

Currently, there are only three phase III studies evaluating this strategy. The NADIM-Adjuvant study from the Spanish Lung Cancer Group (SLCG) randomises completely resected IB-IIIA NSCLC patients to four cycles of carboplatin + paclitaxel + nivolumab at a dosage of 360 mg iv q3 weeks versus carboplatin + paclitaxel, followed by adjuvant nivolumab 480 mg iv q4 weeks × 6 months vs. observation. On the other hand, the MERMAID-1 study compares the efficacy of adjuvant durvalumab in combination with standard ChT versus adjuvant standard ChT + placebo in completely resected patients with stage II-III NSCLC. The primary objective of these studies is DFS. The other study is ALCHEMIST Chemo-IO (Arm C), one of the studies from the ALCHEMIST group for patients without driver mutations in the molecular screening which, in one of the branches, evaluates the efficacy of four cycles of tri-weekly pembrolizumab in combination with platinum-based doublets, followed by another 13 cycles of pembrolizumab in monotherapy in IB-IIIA NSCLC [44]. The primary aim of this last study is DFS and OS.

### 4.3. Adjuvant Treatment with Double Immunotherapy

The studies evaluating this strategy are phase II and assess the strategy of double inhibition with anti-PD-L1 and anti-CTLA-4 ICIs with adjuvant durvalumab and tremelimumab, such as the NCT04625699 study, or the combination of ICI + vaccine, as in the NCT04267237 study (atezolizumab and RO7198457).

## 5. Neoadjuvant Strategies in the Age of Immunotherapy

### 5.1. Neoadjuvant Immunotherapy +/− Chemotherapy Strategy

Neoadjuvant immunotherapy can play a fundamental role in increasing complete resection rates (R0) and controlling micrometastases in early stages.

The first immunotherapy strategy explored in neoadjuvant treatment was the use of ICIs in monotherapy, with the CheckMate-159 study being the first published prospective study. It was a phase I study in which safety was the main objective, and included patients diagnosed with stage I-IIIA NSCLC (including 33% stage IIIA, but without specification of how many had N2 involvement) [47]. They received two cycles of nivolumab prior to surgery and obtained a major pathological response (MPR) rate of 45% (the highest of studies with these characteristics) and a 15% pCR, without drug-related deaths or surgical delays. In addition, with a median follow-up of 30 months, the median progression-free survival (PFS) was still not reached, which was 69% at 24 months (95%CI: 51–93) [48]. The phase II study LCMC3, for which preliminary data were presented at the 2019 ASCO annual meeting (with 101 of the 180 expected patients), included 101 patients with stage IB-IIIB NSCLC (39 IIIA and 7 IIIB without specifying N2 involvement), 46% of them stage III, in which two doses of atezolizumab were administered prior to surgery. The primary aim was MPR, achieving 40.5%, as well as 5% pCR [49]. In this case, they do report a single surgery delay due to immune-mediated toxicity (grade 3 pneumonitis). Another similar, more recent study, is ChiCTR-OIC-17013726, a phase IB study that treated patients with stage IA-IIIB NSCLC with two doses of neoadjuvant sintilimab, with an MPR of 40.5%. Though the primary aim was safety, similar to previous studies, one case of immune-mediated grade 5 pneumonitis can be highlighted [50]. Some of these patients also received adjuvant treatment with chemotherapy, sintilimab or a combination of both, according to their initial response to the drug.

Another option that has been explored is combined double immunotherapy with nivolumab and ipilimumab in the phase II study NEOSTAR, also presented at the ASCO meeting 2019, in patients with stage I-IIIA NSCLC (20% IIIA patients, with single N2 involvement). It was a two-arm study: three doses of neoadjuvant nivolumab vs. adding a single dose of ipilimumab to the former. The primary endpoint was MPR, which was 24%, with a pCR of 15%. Regarding the toxicity of the ICI combo, only one case of grade 3 diarrhoea was reported, and the remaining immune-mediated adverse events occurred in the nivolumab group, including one grade 5 pneumonitis [51].

Nevertheless, the best MPR and pCR results were achieved with the synergistic effect of neoadjuvant immunotherapy and ChT combination, with this being superior to neoadjuvant immunotherapy alone. TOP1201 phase II study analysed the immunomodulatory potential of the anti-CTLA4 ICI ipilimumab in circulating T cells of Stage II-IIIA NSCLC resectable patients (79% Stage IIIA). It evaluated the safety and plausibility of three cycles of platinum-based doublet, with paclitaxel and ipilimumab in cycles 2 and 3, achieving a pCR of 15.4%. The MPR has not been reported and no increase in toxicity or perioperative mortality was observed [52]. On the other hand, the phase II study from Shu et al. [53], which included patients with stages IB-IIIA NSCLC (77% IIIA), used four cycles of neoadjuvant atezolizumab with nab-paclitaxel and carboplatin, achieving an MPR (primary endpoint) of 56.7%, and a pCR of 33.3%. Lastly, the phase II study NADIM from SLCG is the study that has obtained the best results to date. It included potentially resectable stage IIIA patients and is the only study whose primary aim was PFS at 24 months [54]. It evaluated the efficacy and tolerability of three cycles of neoadjuvant ChT with nivolumab, carboplatin and paclitaxel to, after intervention, continue with adjuvant nivolumab for one year, achieving an MPR of 83% and a pCR of 59%. The most commonly observed grade 3-4 toxicity in these two last studies was hematotoxicity. An increase in immune-mediated toxicity was not specifically described. The phase II randomised NADIM II study is ongoing and will try to endorse the results of NADIM study. 

Despite the obtained results being spectacular and highly promising, confirmatory data from phase III studies is lacking as well as long-term data regarding OS. Remaining work to be carried out includes optimising patient selection to minimise risks during the intervention and delving into knowledge about biomarkers that can help us to select patients better.

In view of the extraordinary results of neoadjuvant chemo-immunotherapy in potentially resectable NSCLC with an up to 90.2% downstaging rate and an MPR rate of 83% in the NADIM study, the induction of chemo-immunotherapy could be an interesting option for unresectable selected stage III-N2 NSCLC patients, and if the tumour stage is downgraded, surgery could be re-evaluated. Nevertheless, this strategy should wait until positive results of randomised phase III trials are achieved.

See a summary of commented neoadjuvant studies with immunotherapy +/- chemotherapy in Table 2.

### 5.2. Neoadjuvant Radiotherapy + Immunotherapy Strategy

RT induces immunological changes in tumour cells (Table 3) and could act synergistically with immunotherapy through the release of tumour antigens and the modulation of the tumour microenvironment, generating a better local as well as systemic immune response (abscopal effect) [55,56]. Such combination has shown positive results in preclinical and clinical studies.

This strategy of immunotherapy in combination with radical RT is not yet backed by solid clinical data, but it is being evaluated in several trials. In the setting for neoadjuvant treatment in stage III-N2 NSCLC, five ongoing trials stand out (Table 4).

The phase II trial NCT03237377 is evaluating the safety and efficacy of neoadjuvant radioimmunotherapy in stage III NSCLC. Patients receive durvalumab and RT or durvalumab + tremelimumab and RT in a non-randomised manner. The multi-centre, phase II trial SAKK 16/18 (NCT04245514) is evaluating the efficacy and safety of the concomitant administration of neoadjuvant CRT and durvalumab prior to surgery in locally advanced NSCLC. In the phase I trial CASE 4516 (NCT02987998), 20 patients with stage IIIA NSCLC received neoadjuvant CRT plus pembrolizumab followed by surgery and consolidation pembrolizumab. The phase II trial NCT02904954 analyses the overall response rate (ORR) and PFS of two groups of patients with stage I-III NSCLC who are randomised to receive two doses of neoadjuvant durvalumab with or without SBRT. Lastly, the phase II trial NCT03871153 is analysing the results of concomitant CRT plus durvalumab followed by surgery and adjuvant durvalumab in stage III-N2 NSCLC.

These trials will likely establish the benefit and safety of radioimmunotherapy ± ChT in the neoadjuvant treatment of stage III-N2 NSCLC.

### 5.3. Role of Minimal Residual Disease in Neoadjuvant and Adjuvant Setting

Minimal residual disease (MRD), measured by the detection of circulating tumour DNA (ctDNA) in liquid biopsy, has an established prognostic role in the treatment of patients with advanced NSCLC and several potential applications in different settings throughout the management of patients with NSCLC [64]. Some ongoing studies are trying to elucidate whether MRD assessment has utility in adjuvant setting. The MERMAID-1 phase III trial (NCT04385368) is evaluating DFS in MRD-positive patients with resected stage II-III NSCLC treated with adjuvant durvalumab plus chemotherapy followed by durvalumab or placebo plus chemotherapy followed by placebo for up to 12 months [65]. On the other hand, the MERMAID-2 phase III trial (NCT04642469) is evaluating the efficacy of durvalumab vs. placebo in patients with MRD-positive resected stage II-III NSCLC, while those patients without MRD will only undergo surveillance. The role of MRD in neoadjuvant/adjuvant setting is also being studied as part of hypothesis-generating translational research in the NADIM study seeing whether pre-treatment ctDNA levels may predict survival in neoadjuvant-treated NSCLC patients.

## 6. Combination of Immunotherapy and Definitive Radiotherapy +/− Chemotherapy Strategy

The strategy of immunotherapy in combination with concomitant RT +/− ChT started out by evaluating the combination of RT with different vaccines and interferons, but what truly changed the history of unresectable III-N2 disease was the addition of ICIs with anti-PD1/PD-L1, improving OS data that had not been improved in the previous decades. First, sequential immunotherapy after radiotherapy +/- chemotherapy was evaluated (Table 5).

Regarding the ICIs with radical RT combination, the greatest success was that of the phase III, randomised, placebo-controlled PACIFIC study, which demonstrated a significant increase in OS and PFS by adding consolidation durvalumab treatment, an anti-PDL-L1 antibody, after radical CRT in patients with unresectable stage III NSCLC. With a median follow-up time of 34.2 months, the 4-year OS for the durvalumab group was 49.6% vs. 36.3% in the placebo group, and 4-year PFS was 35.3% vs. 19.5%, respectively. In the last update on the study results, the median OS was 47.5 vs. 29.1 months (HR 0.71; 95%CI, 0.57–0.88) and the median PFS was 17.2 vs. 5.6 months (HR 0.55; 95%CI, 0.44–0.67) [66]. Regarding toxicity, the grade 3 or 4 adverse events were 29.9% in patients with durvalumab vs. 26.1% with placebo, and the most common grade 3 or 4 event was pneumonia, 4.4 vs. 3.8, respectively [67]. Following these results, the Food and Drug Administration (FDA) and Europe Medicine Agency (EMA) approved the use of consolidation durvalumab following radical CRT as standard treatment, although the EMA limited its indication to tumours with PD-L1 expression (≥1%) based on a *post hoc* study of subgroups in which tumours without PD-L1 expression did not receive a significant benefit to survival, a situation that is highly debated within the scientific community. This strategy is not a choice for patients with potentially resectable III-N2 NSCLC, although given the positive results, the argument has been made for its potential benefit to some of these patients compared with the multimodality treatment strategy with resection surgery.

The phase II study LUN 14-179 was recently published, with consolidation pembrolizumab following CRT, which shows an increased time to metastatic disease or death, PFS and OS compared to historical controls with CRT alone (44.6% and 22.4 months and 35.8 months, respectively). The rate of grade 3 to 5 pneumonitis was similar to that reported for exclusive CRT [68]. The nivolumab/ipilimumab combination vs. nivolumab after CRT is also being studied, with increased grade 3 toxicity in the combination arm, though recruitment continues [69].

Subsequently, the immunotherapy and RT combination has been evaluated, moving towards administering it concomitantly (Table 6). The first studies used interferon alpha and beta, with increased toxicity and no improvement in OS [70,71,72,73]. A meta-analysis analysing the reinfusion of cytokine-induced killer cells (CIKs), lymphocyte-activated killer cell (LAK) or tumour-infiltrating lymphocyte (TIL) with CRT found a statistically significant improvement in 2-year OS (OR 2.45; 95%CI, 1.6–3.75, *p* < 0.001) [74]. A second meta-analysis on the efficacy of CIKs with RT also found improvement in 3-year OS (OR 1.66; 95%CI, 1.2–2.29) [75]. However, following the excellent results of the PACIFIC study, major research efforts have focused on documenting the efficacy of the concomitant use of anti PD(L)-1 with CRT. The phase II study DETERRED evaluates the safety and efficacy of adding anti-PD-L1, atezolizumab, successive to CRT as maintenance versus concomitantly and as maintenance. For the moment, the only toxicity data that have been reported has been similar in the two arms [76]. Likewise, the NICOLAS trial evaluates the safety of administering anti PD-1 nivolumab with CRT. With a follow-up of 3 months following the completion of RT for 21 patients, no grade 3 pneumonitis has been observed. The preliminary results show a 12-month PFS of 54% (95%CI, 41–65) with a median PFS of 12.4 months and 12-month OS of 79% (95%CI, 68–87), while the median OS has not yet been reached [77]. Presented at the ASCO meeting 2020, the preliminary results, with recruitment yet to be completed, of the phase II, non-randomised trial KEYNOTE-799, with pembrolizumab concomitant to CRT, show that the overall response rate (ORR) was 67% in patients with carboplatin and paclitaxel-based ChT, and 56.6% with platinum and pemetrexed. The percentage of grade 5 pneumonitis was 3.5% with carboplatin + paclitaxel [78]. Likewise, results have been presented for atezolizumab prior and subsequent to CRT treatment, with an ORR of 82.4% for PD-L1 negative patients and 90.9% for PD-L1 positive. Severe toxicity was 21% [79].

There are even fewer data about the role of immunotherapy for resectable patients. A phase I study tried to analyse the combination of platinum-based ChT, etoposide and pembrolizumab with RT, 45 Gy in 25 fractions, followed by surgery and 6 months of consolidation pembrolizumab. The primary aim was safety and feasibility, but with nine patients included, the study ended due to a high rate of grade 3 toxicity and two deaths, despite having achieved a pCR of 67% [80].

## 7. Perioperative Systemic Treatment in N2 NSCLC with Targetable Mutations

The development of precision medicine in lung cancer, led by tyrosine kinase inhibitors (TKIs) versus different molecular oncogenic-driver alterations, such as the *EGFR* mutation or the *ALK* fusion, has transformed the paradigm for advanced disease, and they are the standard treatment in patients with such alterations, with different therapeutic sequences based on the routinely available TKIs. The addition of targeted therapy to the perioperative phase has already been shown to be effective, improving DFS data in patients with *EGFR* mutation, with a significant increase in DFS with gefitinib and osimertinib [81,82]. As neoadjuvant treatment, the patients receive between one and three months of TKI treatment prior to surgery, while in the case of adjuvant treatment, the duration usually lasted between one and three years after surgery [83].

There is little available evidence on immunotherapy in the perioperative context in these populations with molecular alterations. The first studies were designed prior to knowledge of the limited impact of immunotherapy on these populations [84]; therefore, these patients were included (molecular testing prior to inclusion was not a requirement). We hope that safety and efficacy data are progressively reported, which will allow us to know if there is or is not a place for immunotherapy in this context. Nevertheless, the evidence will be limited to these subgroup analyses given that, at the current moment, the *EGFR* mutation or an *ALK* infusion are considered exclusion criteria for the majority of current studies.

## 8. Surgery Issues in Patients Treated with Perioperative Immunotherapy

The published clinical trials with immunotherapy induction schedules or combined immuno-chemotherapy strategies are showing higher pathological response rates and never before seen pathological complete responses, which can benefit surgical indications in patients with stage IIIA-N2. It is worth noting the importance of strictly observing the criteria for complete resection in lung cancer lung resection [85] in the patients included in the studies, ensuring the use of uniform and strict procedures for evaluating the pathological response.

Given that downstaging after induction therapy is a better prognostic factor, mediastinal re-evaluation is needed to indicate intervention, as only cyN0-N1 patients are surgical candidates in most cases. The reduction in diagnostic yield is known in mediastinal restaging following induction with chemo- or chemo-radiotherapy, in both non-invasive (positron emission tomography–computed tomography (PET-CT)) [86] and invasive testing (endobronquial ultrasound–transbronquial needle aspiration (EBUS-TBNA)/endoscopic ultrasound–fine needle aspiration (EUS-TNA) and mediastinoscopy) [87,88,89] and it is highly likely that the same occurs in patients induced with immuno- or chemo-immunotherapy. As some response patterns can falsify the results of a PET-CT (pseudoprogression), the regular criteria for evaluating response with this technique are not valid within this context; new assessment criteria have been reported, pending validation, that help correct the classification of response in these patients [90]. Therefore, due to the fact that these patterns that can produce false positives, patients should not be ruled out of surgery due to a PET-CT that suggests a lack of response or progression. Rather, an invasive technique should be adopted in the event of uncertainty, assuming a lower diagnostic yield for the same. Information on the capacity of EBUS-TBNA to differentiate a “sarcoid”-like pseudoprogression from true involvement has been published. Regarding the mediastinoscopy, it may be complex and offer low yield due to inflammation and fibrosis, but is particularly recommended in the event that initial staging was performed [91] with EBUS-TBNA.

Once surgical intervention is indicated, surgery may be performed in just 1–2 weeks if induction is carried out exclusively with immunotherapy. Nevertheless, if schedules involving chemotherapy are used, it must be delayed by the regular 4–6 weeks.

Though little data have been published on the surgical results of trials using immunotherapy as induction treatment, surgery is possible and safe, achieving complete resection in 100% of the patients. The reported postoperative mortality is 0% [91], with a complication rate of between 29 and 50% [92,93] and the most common mild complications being atrial fibrillation and persistent air leak [92]. These results are similar to those published with regular induction strategies [19,94,95].

The surgeon should expect a different and technically demanding surgery due to the presence of multiple inflammatory lymph nodes. Thoracotomy is the most commonly used access; however, minimally invasive surgery is always indicated as long as oncological resection is ensured, as a high rate of conversion to open surgery has been published (23–54%) [92].

## 9. Conclusions and Future Perspectives

Considering lung cancer is the most common type of cancer globally and the largest source of mortality due to cancer, and the fact that its survival rates are still low even in early stages, with 36% and 26% survival at 5 years for stage IIIA and IIIB, respectively, improving management of resectable III-N2 NSCLC is a true necessity and a priority for the scientific community in this field, as it represents a significant health care issue.

Stage III-N2 NSCLC cases continue to be a heterogeneous group with various potential therapeutic approaches, but all of them entail multidisciplinary management for which surgery is indicated in some cases and not in others. In general, patients with N2 involvement are immediately considered unresectable, barring some exceptions for the non-bulky involvement of single N2 lymph node, but for those potentially resectable III-N2 NSCLC cases, the current standard treatment is starting neoadjuvant ChT with three cycles of platinum-based doublet followed by restaging to assess salvage surgery and subsequent assessment for adjuvant ChT. On the other hand, immunotherapy is already included in the therapeutic arsenal against unresectable stage III and advanced NSCLC, and the promising data obtained in phase II trials on neoadjuvant treatment lead us to think that the addition of anti-PD1/PD-L1 ICIs to the management of III-N2 NSCLC will be the next step to improving OS data and improving the cure rate in these patients.

For those patients operated on due to unexpected III-N2 NSCLC, there are adjuvant therapy studies with anti-PD1/PD-L1 ICIs in monotherapy and in combination with ChT, as well as one study with anti-ILß canakinumab antibodies, with a duration of between 6 months and 1 year; however, we still do not have mature data. In patients with potentially resectable III-N2 NSCLC, we already have data from phase II trials on neoadjuvant treatment with ICIs as monotherapy, double inhibition with ICIs, and a combination of ICIs and ChT. To date, these have shown very promising results for pCR and MPR as never seen before, and for DFS, pending confirmation that these are surrogate variables for OS, and the confirmatory results from the randomised and phase III trials.

There are also various trials evaluating the safety and benefits of radioimmunotherapy ChT +/− neoadjuvant treatment, and in those with unresectable III NSCLC, CRT followed by consolidation durvalumab is a standard treatment. Moving immunotherapy forward to the concomitant phase with RT is also being evaluated, and in light of its positive results, one of the questions will be if patients with resectable III-N2 could also benefit from this strategy, in which surgery is not involved.

In III-N2 patients with driver mutations, the available evidence with perioperative immunotherapy is scarce and, to date, the route which tends to be followed is that of perioperative targeted therapy.

In terms of surgical treatment, we must bear in mind diverse aspects such as the different response patterns to immunotherapy that can produce false positives and the complexity of hilar fibrosis; however, according to the published data, it seems that surgery is safe and possible following use of neoadjuvant immunotherapy.

In the very near future, we will see immunotherapy integrated into the perioperative management of resectable III-N2 NSCLC. The following steps will lead to an even better improvement in cure rate with new combinations: to define optimal treatment duration, to work in the development of biomarkers and to determine the role of immunotherapy in cases of relapse or progression.

## Figures and Tables

**Table 1 cancers-13-04811-t001:** Studies evaluating adjuvant immunotherapy in Stage III-N2 NSCLC.

Study	Strategy	Phase	Patients	N	Experimental Arm	Control Arm	Primary Endpoint	Study End
PEARLS/KEYNOTE-091	IO monotherapy after SOC ChT	III	IB (≥4 cm)-IIIA	1080	Pembrolizumab 200 mg q3w IV 1 year	Placebo 1 year	DFS	2024
BR31/LINC	IO monotherapy after SOC ChT	III	IB (≥4 cm)-IIIA	1360	Durvalumab 10 mg/kg q2w IV 6 mo 20 mg/kg q4w IV 6 mo	Placebo 1 year	DFS in PD-L1+ and ITT	2024
ANVIL	IO monotherapy after SOC ChT	III	IB (≥4 cm)-IIIA	903	Nivolumab 240 mg qw IV 1 year	Observation	DFS OS	2024
IMpower 010	IO monotherapy after SOC ChT	III	IB (≥4 cm)-IIIA	1280	Atezolizumab 1.200 mg q3w IV 1 year	Observation	DFS in II-IIIA, II-IIIA PD-L1+, and ITT	2027
ALCHEMIST-IO arm B	IO monotherapy after SOC ChT	III	IB (≥4 cm)-IIIA	1263	Pembrolizumab 200 mg q3w IV 17 cycles	Arm A: observation Arm C: ChT + pembro × 4 pembro × 13	DFS OS	2024
CANOPY-A	IO monotherapy after SOC ChT	III	II-IIIA, IIIB (T > 5 cm y N2)	1500	Canakinumab 200 mg q3 sc 1 year	Placebo 1 year	DFS	2027
NADIM-Adjuvant	IO + ChT	III	IB (≥4 cm)-IIIA	210	Carboplatin + paclitaxel + Nivolumab 360 mg IV q3w × 4 cycles, then Nivolumab 480 mg IV q4w 6 m	Carboplatin + paclitaxel and observation	DFS	2028
ALCHEMIST Chemo-IO (ACCIO) Arm C	IO + ChT	III	IB (≥4 cm)-IIIA	1263	Pembrolizumab 200 mg q3w + standard ChT × 4 cycles pembro IV 13 cycles	Arm A: standard ChT + observation Arm B: pembrolizumab 17 cycles	DFS OS	2024
MERMAID-1	IO + ChT	III	II-III	332	Durvalumab + standard ChT	Placebo + standard ChT	DFS in MRD+	2026
NCT04625699	Double IO	II	II-IIIB with ctDNA +	15	Durvalumab 1500 mg q4w IV × 4 cycles + Tremelimumab 300 mg q56 days IV × 2 cycles	-	Feasibility	2022
NCT04267237	Double IO	II	II-III	NS	Atezolizumab 1680 mg q4w IV + RO7198457 q4w IV × 12 cycles	Atezolizumab 1680 mg q4w IV × 12 cycles	DFS	2025

IO = immunotherapy; ChT = chemotherapy; NS = not specified; IV = intravenous; w = week; DFS = disease-free survival; OS = overall survival; MRD = minimal residual disease; ITT = intention to treat population; ctDNA = circulating tumour DNA; SOC = standard of care; sc = subcutaneous.

**Table 2 cancers-13-04811-t002:** Neoadjuvant studies with immunotherapy +/− chemotherapy in stage III-N2 NSCLC.

Study	Phase	Primary Endpoint	NSCLC Stage (% Stage III)	R0 Patients	Neo-IO	Neo-ChT	iRAE ≥ 3	pCR Rate	MPR Rate
CheckMate-159 [47]	I	Safety	I-IIIA (33%)	91%	Nivolumab 3 mg/kg/2 w × 2 cycles	None	5%	15%	18%
LCMC3 [49]	II	MPR	IB-IIIB (46%)	101	Atezolizumab 1200 mg, D1 and D22	None	6%	5%	40.5%
ChiCTR-OIC-17013726 [50]	Ib	Safety	IA-IIIB (45%)	93%	Sintilimab 200 mg q32 × 2 cycles	None	10%	16.2%	24%
NEOSTAR [51]	II	MPR	I-IIIA-single N2 (20%)	44 (23 N, 22 N + I)	N 3 mg/kg D1,15,29 or N + I 1 mg/kg D1	None	NR	15%	25% (7 N, 4 N + I)
TOPT1201 [52]	II	T cells ^	II-IIIA (75%)	54%	Neoadjuvant Ipilimumab 10 mg/kg q3w cycles 2 and 3 + Adjuvant I q3w × 2 cycles	Paclitaxel 175 mg/m^2^ + cisplatin 75 mg/m^2^ or carboplatin AUC6 × 3 cycles	NR	15.4%	NR
Shu et al. [53]	II	MPR	IB-IIIA (77%)	87%	Atezolizumab 1200 mg, q3w × 4 cycles	Nab-paclitaxel 100 mg/m^2^ D1,8,15 + carboplatin AUC5 D1/21d	NR	33.3%	56.7%
NADIM [54]	II	24-m-PFS	IIIA (100%)	89%	Neoadjuvant N 360 mg q3w × 3 cycles + adjuvant N 1 y	Paclitaxel 200 mg/m^2^ + carboplatin AUC6	NR	59%	83%

^ Percentage of subjects with detectable circulating T cells after treatment. MPR = major pathological response (<10% viable tumor). pCR = pathological complete response. NS = not specified, m = months, iRAEs = immune-related adverse events, N = nivolumab; I = ipilimumab. DRAEs= drug-related adverse events; PFS = progression-free survival; w = weeks; D = day; d = days; AUC = area under curve.

**Table 3 cancers-13-04811-t003:** Potential immunomodulatory mechanisms of radiotherapy.

Up-regulation of MHC Class I antigens [57].
2. Proinflammatory signals release increase (citokines…) [58].
3. Boost of a polyclonal T-cell response [59].
4. Enhancement of CD8 T cell activation [58].
5. Increased antigen presentation and dendritic cells priming [60].
6. Modulation of the tumour microenvironment [61].
7. IFN gamma response [62].
8. Up-regulation of PD-L1 expression in the tumour cell [63].

MHC: major hystocompatibility complex.

**Table 4 cancers-13-04811-t004:** Ongoing radio-immunotherapy studies in stage III NSCLC.

NCT Number/Trial Name	Study Phase	Patients	IO Agent	Trial Design	RT Dose	RT and IO Timing	Status
NCT03237377	II Single arm	Resectable Stage IIIA NSCLC	Durvalumab ± tremelimumab	Neoadjuvant IO + RT, followed by surgery	45 Gy/25 fx	Concurrent	Recruiting
NCT04245514	II Single arm with 3 radiotherapy cohorts	Resectable Stage IIIA NSCLC	Durvalumab	Neoadjuvant IO + RT followed by surgery	Allocated in a 1:1:1 ratio: Arm A: 20 × 2 Gy Arm B: 5 × 5 Gy Arm C: 3 × 8 Gy	Concurrent	Recruiting
NCT02987998	I	Resectable Stage IIIA NSCLC	Pembrolizumab	Neoadjuvant chemoRT (cisplatin-etoposide) + IO, followed by surgery, followed by consolidation IO	45 Gy/25 fx	Concurrent	Active, not recruiting
NCT02904954	II Randomised	Resectable Stage I-IIIA NSCLC	Durvalumab	Neoadjuvant IO ± SBRT, followed by surgery, followed by postoperative maintenance IO	SBRT to 24 Gy/3 fx	Concurrent	Active, not recruiting
NCT03871153	II Single arm	Resectables Stage IIIA NSCLC	Durvalumab	Neoadjuvant chemoRT (carboplatin–paclitaxel) + IO followed by surgery	45–61.2 Gy 1.8–2.0 Gy per day		Recruiting

IO = immunotherapy; RT = radiotherapy; NSCLC = non-small-cell lung cancer; Gy = grays; fx = fractions; SBRT = stereotactic body radiation therapy.

**Table 5 cancers-13-04811-t005:** Studies evaluating sequential immunotherapy with radiotherapy in stage III NSCLC.

Author	Phase	Patients	Immunotherapy	RT/ChT	Primary Outcome
Brunsvig et al., 2011	II	24	GV-1001 after CRT	66 Gy 30 fx/weekly docetaxel	No serious AE 66.6% responders with mOS 19 m. Non responders with mOS 3.5 m.
START Butts et al., 2014	III	1239	Tecemotide after concurrent or sequential CRT vs. placebo	50 Gy/platin-based ChT × ≥2 cycles	mOS 25 vs. 22.3 m (HR 0.54, 95%CI 0.301–0.999; *p* = ns)
Katakami et al., 2017	I/II	172	Tecemotide after concurrent CRT vs. placebo	66 Gy 33 fx/carboplatin–paclitaxel	6% G5 toxicity mOS: 32.4 vs. 32.2 m (HR 0.95, 95%CI 0.61–1.48; *p* = 0.83)
PACIFIC Antonia et al., 2017	III	709	Durvalumab after concurrent CRT vs. placebo	54–66 Gy, 27–30 fx/platin based ChT > 2 cycles	G3-4 AE: 29.9% durvalumab and 26.1% placebo; G3-4 pneumonia: 4.4% durvalumab and 3.8% placebo OS: HR = 0.69, 95%CI: 0.55–0.86
LUN 14-179 Durm et al., 2020	II	93	Pembrolizumab after concurrent CRT	59.4–66.6 Gy/cisplatin-etoposide or cisplatin-pemetrexed or carboplatin–paclitaxel	G3–4 AE 4.3%; G5:1.1% mOS: 35.8 m (95%CI, 24.2 to not reached)
Patel et al., 2020	II	33	Tecemotide + Bevacizumab after CRT	66 Gy/33 Gy fx, concurrently with ChT	≥G3 AE in 11 pts, G5 2 pts. mOS 42.7 m (95%CI, 21.7–63.3)

AE = Adverse events; fx = fraction; ns = not significant; mOS = median overall survival; m = months; CRT = chemoradiation; RT = radiotherapy; ChT = chemotherapy; Gy = grays; fx = fractions; mOS = median overall survival; G = grade; HR = hazard ratio; CI = confidence interval.

**Table 6 cancers-13-04811-t006:** Studies evaluating immunotherapy with concurrent CRT in stage III NSCLC.

Study	Phase	Patients	Immunotherapy	RT/ChT	Primary Outcome
Maasilta et al., 1992	II	20	Alpha-INT with RT vs. RT alone	66 Gy, 1.25 Gy/fx tw	Moderate/severe pneumonitis and/or oesophagitis in experimental arm
McDonald et al., 1993	I/II	39	Beta-INF with RT	54–59.4 Gy, 1.8 Gy/fx	ORR: 81% CR: 44% 5-y OS 51% No serious AE
Shaw et al., 1995	I	18	Gamma-INF with RT	60 Gy, 1.5 Gy/fx tw	50% life threatening or fatal AE mOS: 7.8 m
RTOG 93-04 Bradley et al., 2002	III	123	Beta-INF with RT vs. RT	60 Gy, 2 Gy/fx	G3-4 AE higher on beta-INF arm *p* = 0.024 1-y OS: 44% vs. 42% (*p* = ns)
DETERRED Lin et al., 2018	II	40	Atezolizumab + CRT	60–66 Gy, 30–33 fx, concurrently with ChT	≥G3 atezo-related toxicity in 6 pts 30%; G5 fistula (*n* = 1) 5%. G3 radiation pneumonitis (*n* = 1)
NICOLAS Peters et al., 2019	IA/II	79	Nivolumab with CRT	66 Gy/33 fx, concurrently with ChT	No ≥G3 post-RT pneumonitis 1-y PFS 50% and 1-y OS 79%
KEYNOTE 779 Jabbour et al., 2021	II	185	Pembrolizumab with CRT	60 Gy, 2 Gy/fx/carboplatin–paclitaxel (A) or cisplatin-pemetrexed (B)	G ≧ 3:AE A: 64.3; B: 41, pneumonitis > 3 A: 8%, B: 5.5%
2020 AFT-16 Ross et al.	II	64	Atezolizumab before and after CRT	60 Gy, 2 Gy/fx+ carboplatin–paclitaxel	AE ≧ 3: 20% 12-week-DCR: 77.4%
Lemmon et al., 2020	I	9	Pembrolizumab with CRT + Surgery	45 Gy, 1.8 fx with cisplatin-etoposide	pCR 67% 2 G5 AE Trial halted

RT = radiotherapy; ChT = chemotherapy; CRT = chemoradiotherapy; AE = adverse event; INF = interferon; ns = not significant; Gy = grays; fx = fraction-s; ORR = overall response rate; CR = complete response; DCR = disease control rate; pCR = pathologic complete response; OS = overall survival; PFS = progression-free survival; tw = twice; G = grade; m = months.

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
