# Peer review of "Management of Resectable Stage III-N2 Non-Small-Cell Lung Cancer (NSCLC) in the Age of Immunotherapy"

_cancers, 2021, doi:10.3390/cancers13194811_

Round 1

Reviewer 1 Report

The authors provide a well written, comprehensive review of the management of resectable stage III-N2 Non-small-cell lung cancer (NSCLC) in the age of immunotherapy. This is an extremely relevant issue since the use of Iimmunotherapy in this setting might contribute to increased cure rates.

I have some minor comments:

  • In the adjuvant setting, the evolving role of biomarkers of MRD such as ctDNA (eg. MERMAID) needs to be discussed.
  • Chemo-IO- combinations are effective in downstaging and may therefore make primarily irresectable NSCLC resectable. This important scenario should included in the review since real world evidence in favour of this approach is accumulating.

Author Response

September 20, 2021

Dear Editor,

Please find enclosed the edited manuscript in Word format

Title: Management of resectable stage III-N2 Non-small-cell lung cancer (NSCLC) in the age of immunotherapy

1) The manuscript has been updated

2) The language has been polished

3) Modifications were done according to the reviewer suggestions as follows:

Reviewer #1

Comment 1: In the adjuvant setting, the evolving role of biomarkers of MRD such as ctDNA (eg. MERMAID) needs to be discussed.

Response: We added a new sub-section (5.2) at the end of the section talking about immunotherapy in neoadjuvant setting. In this new sub-section we discuss the evolving role of MRD measured as ctDNA in adjuvant and neoadjuvant studies.

Comment 2: Chemo-IO- combinations are effective in downstaging and may therefore make primarily irresectable NSCLC resectable. This important scenario should included in the review since real world evidence in favour of this approach is accumulating.

Response: We added a new paragraph in the end of 5.1 section talking about the potential approach of using neoadjuvant chemo-IO combo in primarily unresectable NSCLC.

Reviewer #2

Comment 1: Line 184: It is not clear what is “standard adjuvant treatment” in this case. From what stated in the text, it seems that all studies cited in this section aim to evaluate adjuvant immunotherapy vs placebo or observation. Please check this section and clarify this point.

Response: All studies allowed to receive standard adjuvant treatment with chemotherapy as we explained in the introduction. To make it more clear, I have specified it on “Strategy” column for each study on Table 1.

Comment 2: Tables are very useful, maybe the format should be harmonized

Response: We have harmonized, improved and updated all tables.

Comment 3: Table 1: ALCHEMIST-IO arm B is reported twice

Response: We have specified ALCHEMIST-IO arm B in one row, and ALCHEMIST Chemo-IO arm C in the other row.

Comment 4: Table 3: Please add references supporting the immunomodulatory mechanisms of radiotherapy

Response: We have added references supporting each statement

Comment 5: Line 341: Please define CIK, CD-CIK, LAK or TIL

Response: We have defined all acronyms.

Comment 6: Title of section 7. Perioperative systemic treatment in oncogenic-driven N2 NSCLC: I think the Authors mean oncogene-driven. However, it would be more correct to speak about NSCLC with targetable mutations, since many cancers are driven by oncogenes which are not targetable yet and are treated like non-mutated cancers.

Response: We have changed the title of section 7 as suggested.

Comment 7: Limited English revision is needed. Some suggestions are as follows: Line 150: “possible” maybe changed into “possibly”; Line 176: “…results are no available.” Should be changed into “…results are not available.”; Title of Table 5: “Secuential” should be changed into “sequential”; Lines 470-473: Please change “to improving” with “to improve”, “to defining” with “to define” and “to determining” with “to determine”

Response: We have corrected all these English mistakes and imperfections as suggested.

Comment 8: Please also note the following sentences I found difficult to follow or unclear: Lines 31-32; Lines 41-43; Lines 160-162; Lines 250-255

Response: We have rewritten this sentences/paragraphs to make them more clear.

Reviewer #3

Comment 1: The Authors should provide the extensive forms for all acronyms through the text when they first appear. Please add the expand forms for all acronyms in tables.

Response: We defined all acronyms through the manuscript and we also added the expand forms for all acronyms in tables. Tables have been updated.

Overall, we would like to thank the reviewer remarks which have help us to improve substantially our manuscript.

4) References have been updated. We have updated reference 7 and addes some refferences: 39, 46, 57, 58 ,59, 60, ,61, 62, 63, 64, 65, 96, ,97.

Thank you again for publishing our manuscript in Cancers.

Sincerely yours,

Xabier Mielgo-Rubio, MD

Department of Medical Oncology

Hospital Universitario Fundación Alcorcón, Madrid, Spain

Calle Budapest 1, Alcorcón, 28922, Madrid, Spain

E-mail: [email protected] /[email protected]

Reviewer 2 Report

This review article from Mielgo-Rubio and colleagues aims to summarize and discuss the advances in the use of immunotherapy in the setting of resectable stage III-N2 NSCLC.

Although the range of topics covered is very limited, the review is well-written and exhaustive, discussing both state-of-the-art clinical practice and the most recent clinical trials data.

I have only minor issues:

  • Line 184: It is not clear what is “standard adjuvant treatment” in this case. From what stated in the text, it seems that all studies cited in this section aim to evaluate adjuvant immunotherapy vs placebo or observation. Please check this section and clarify this point.
  • Tables are very useful, maybe the format should be harmonized
  • Table 1: ALCHEMIST-IO arm B is reported twice
  • Table 3: Please add references supporting the immunomodulatory mechanisms of radiotherapy
  • Line 341: Please define CIK, CD-CIK, LAK or TIL
  • Title of section 7. Perioperative systemic treatment in oncogenic-driven N2 NSCLC: I think the Authors mean oncogene-driven. However, it would be more correct to speak about NSCLC with targetable mutations, since many cancers are driven by oncogenes which are not targetable yet and are treated like non-mutated cancers.

Limited English revision is needed. Some suggestions are as follows:

  • Line 150: “possible” maybe changed into “possibly”
  • Line 176: “…results are no available.” Should be changed into “…results are not available.”
  • Title of Table 5: “Secuential” should be changed into “sequential”
  • Lines 470-473: Please change “to improving” with “to improve”, “to defining” with “to define” and “to determining” with “to determine”

Please also note the following sentences I found difficult to follow or unclear:

  • Lines 31-32
  • Lines 41-43
  • Lines 160-162
  • Lines 250-255

Author Response

(The authors gave the same response as above.)

Reviewer 3 Report

The manuscript entitled "Management of resectable stage III-N2 Non-small-cell lung 2 cancer (NSCLC) in the age of immunotherapy" highllighted all therapeutic approaches to stage III-N2 NSCLC, analysing both completed and ongoing studies that evaluate the addition of immunotherapy with or without chemotherapy and/or radiotherapy.

  • The Authors should provide the extensive forms for all acronyms through the text when they first appear.
  • Please add the expand forms for all acronyms in tables.

Author Response

(The authors gave the same response as above.)
